# Community Women’s Lifestyle and Eating Disorders in the Era of COVID-19 Pandemic: A 15-Year Follow-Up Study

**DOI:** 10.3390/nu15071676

**Published:** 2023-03-30

**Authors:** Nasim Foroughi, Phillipa Hay, Haider Mannan

**Affiliations:** Translational Health Research Institute, School of Medicine, Western Sydney University, Locked Bag 1797, Penrith, NSW 2751, Australia

**Keywords:** eating disorders, COVID-19, mental health, health care, health literacy, eating disorder-related Medicare item

## Abstract

Most studies suggest that COVID-19 has adversely affected the quality of life and mental health, including eating disorders. However, studies have yet to examine longitudinally the impact of COVID-19 on eating disorder symptomatic individuals. This study aims to examine longitudinally the impact of the COVID-19 pandemic on the lifestyle and eating disorder symptoms of a symptomatic group of community-dwelling women. These women (n = 171) were enrolled in a longitudinal study, completed a COVID-19 modular self-report (post or Qualtrics, 2020/21), and participated in the current study. This study examined a 15th year follow-up. In 2020, 40% were tested for COVID-19. Of these, 87% had negative results; 5.3% self-isolated at home; 20.5% stopped working/studying in person; 28% continued online work/study; and 28% stopped work/studying in person. The pandemic affected sporting activities, music, and club activities (32.7% discontinued); 38% socialized in person; 16% socialized online; and 10% completely stopped socializing. Findings showed that the respondents showed no significant changes in levels of psychological distress (K10: 21.4 ± 9.8 vs. 19.0 ± 7.1, *p* < 0.171), and impaired quality of life (SF12: 50.9 ± 8.0 vs. 48.3 ± 9.5, *p* < 0.055) at 15-year follow-up. Eating disorder symptoms increased over time (EDE-Q global: 2.1 ± 1.4 vs. 2.9 ± 1.4, *p* < 0.013). Observed worsening of eating disorder-related symptoms during the COVID-19 pandemic may be due to interrupted eating patterns, exercise restrictions and the absence of social support. Provision and access to interventions to support those affected by eating disorders are a high priority, especially during these times.

## 1. Introduction

Although some of the previously published studies present with conflicting evidence, such as reduced self-harm behavior and suicide [1], most studies suggest that the 2019 novel coronavirus (COVID-19) infection has had an overall, substantially negative and significant psychological, financial, economic, and social impact on mental health globally (including anxiety, psychiatric disorders, depression, eating disorders), as well as on overall quality of life [2,3,4,5,6], A survey conducted by Ammar et al. in 2020 revealed that various factors, such as food type, frequency of food consumption, and meal regularity, have been affected by the pandemic [7]. One major impact of COVID-19 on food consumption is a shift in food type. Due to lockdowns and supply chain disruptions, individuals have been forced to alter their dietary choices. As a result, many have resorted to consuming cheaper and more processed foods, which may not provide the necessary nutrients required for optimal health. The increased stress and anxiety brought on by the pandemic have also led to a rise in comfort eating, which often involves consuming unhealthy foods [8]. With the closure of schools and offices, many individuals have been forced to work or study from home, leading to changes in their daily routine. This has resulted in irregular eating patterns, with many individuals skipping meals or consuming them at irregular times. In addition, the increased availability of snacks and food at home has also led to more frequent snacking and overeating [9]. Lastly, with the pandemic and associated restrictions limiting individuals’ ability to engage in physical activity, many have reported decreased appetite and reduced food intake. This has led to concerns about the potential impact on nutritional status and overall health.

Devoe et al. (2023) recently conducted a systematic review of the impact of the COVID-19 pandemic on the physical and mental health of people with eating disorders. Fifty three studies were included in this review, including participants with common eating disorders such as anorexia nervosa, binge eating disorders, bulimia nervosa, and other specified feeding or eating disorders [10]. The results of this review showed that during the COVID-19 pandemic the rate of hospital admission, eating disorder symptoms, depression and anxiety and social isolation increased during the lockdown, as well as body mass index (BMI) [10]. The authors of this systematic review suggested that the reported worsening of eating disorder symptoms in the previous qualitative literature may be explained by the lack of accessibility to treatment and care, disrupted routine and structure, negative impact of social media, and lockdown [10]. The changes reported by the above-mentioned systematic review, however, appear to be specific to age, are diagnostic, and time, for example. improved symptoms after the lockdown period.

It is evident that the stress and uncertainty of COVID-19 has been adversely affecting the prognosis and quality of life (QOL) of those affected by eating disorders. Previous research has suggested that the pandemic may have influenced both the severity of symptoms and the rate of relapse in patients diagnosed with eating disorders [11]. In addition, mass death, increased levels of unemployment, overwhelmed hospitals, and concern about financial and food insecurity contributed to the increased vulnerability to the emotional impact of COVID-19 and the worsening of psychological distress. With restrictions in place, lack of social support and interaction, mandated lockdown and other psychiatric stress factors occurring due to the direct or indirect impact of the coronavirus pandemic, it has been increasingly difficult to cope for those affected by eating disorders. However, these results contrast with the findings of another study by Machado et al. (2020), in which eating disorder symptoms and BMI did not change post-lockdown in patients with anorexia nervosa, binge eating disorders, bulimia nervosa and other specified feeding or eating disorders [12].

In 2020, a nationwide research project with the aim of assessing the coping strategies related to mental health during the COVID-19 pandemic was performed in Australia [6]. This study reported a statistically significant increase in the levels of eating disorders in general, including binge eating disorders, anorexia nervosa and bulimia nervosa. More specifically, meal avoiding, grazing behavior, overeating, loss of control, overeating, and binge eating episodes have been reported during lockdown [13]. There has been interplay between these eating disorders and isolation following infectious disease outbreaks and the subsequent exacerbation of psychological disorders, as well as deterioration in the quality of life. For example, following the SARS (severe acute respiratory syndrome) outbreak in 2003, a study explored the development of psychological disorders following isolation. Approximately one fifth of the participants of this study reported a clear correlation between mental distress following isolation and the exacerbation of restrictive as well as binge eating disorders [14].

The environment created by the novel coronavirus pandemic may similarly contribute to the development of an eating disorder in previously unaffected individuals. This is demonstrated by the fact that there have been studies indicating a strong positive correlation between the Eating Disorders Symptom impact scale (EDSIS) and the General Health Questionnaire (GHQ). Further research needs to be carried out to find or to further establish direct implications of COVID-19 in eating disorders. Understanding these changes can help to inform public health policies and interventions aimed at promoting healthy eating habits during these unprecedented times. Therefore, the main objective of the current study is to further explore longitudinally the effects of the COVID-19 pandemic on mental health related eating disorder symptoms, quality of life, and depression and anxiety in a moderately symptomatic eating disorder sample, more specifically at 15-year follow-up. Further, we aimed to investigate the awareness and the use of newly introduced eating disorder Medicare item numbers by the Australian government in this population.

## 2. Materials and Methods

### 2.1. Study Design

We obtained ethical approval from the Western Sydney University’s Human Research Ethics Committee for this observational analytical study (H13832, August 2020). Participants of a research database established 15 years ago [15] (The Women’s Eating and Health Literacy (WEHL)), who gave consent to be contacted for future studies, were contacted and recruited for the current project. This sample consisted of a combination of young women living in Australian Capital Territory and a sample of higher education university and college students. The participants were recruited 15 years ago (baseline) and have been followed up at year 2, 4, 9, 12, and 15. Surveys were distributed to the participants and were completed online via Qualtrics, or by post. All the participants were requested to provide informed consent for participation in the study.

### 2.2. Assessment Instruments

Depression and anxiety symptoms [16] were assessed via the Kessler-10 (K10) questionnaire, which has been reported to have moderate reliability [17]. An average score indicating positive mental health ranges from 10–15 on this scale [18]. Participant’s physical and mental health [19] was measured via the SF12 questionnaire. This questionnaire has sound reliability and validity, reported previously [20]. The scoring is from 0–100 (lowest to highest health level) with the mean and standard deviation score of 50 ± 10 [19]. The Eating Disorder Examination Questionnaire (EDE-Q) was used to assess eating disorder symptoms over the past 28 days prior to study participation [21]. Subscale scores and a global score are derived and Australian community normative data for these have been published previously [22]. This scale has acceptable internal consistency, test–retest reliability and temporal stability. In addition, we collected information on the personal views of the participants and their experiences of the COVID-19 pandemic. The participants were asked about the use and accessibility of the specific Eating Disorder Medicare item numbers introduced by the Australian government in recent years. We asked three questions of the participants and the answers were yes or no.

### 2.3. Statistical Analysis

We used IBM Statistical Package (SPSS version 23) and SAS (version 9.4) for the analysis of the data in this study. The data was inspected for normality. Chi-square and *T* tests were employed for *p* ≤ 0.05. The quantitative data were expressed as mean and standard error (SE) or median and interquartile range after being statistically and visually tested for normality. Open-ended responses were collated and presented by frequency of response. As missing data were low (<10%), no imputation was made. A linear mixed model (marginal model) was used to estimate the trends in eating disorder and quality of life outcomes longitudinally over time. A test of linear trend was performed using *t* test for *p* < 0.05. A rejection of this test indicated that there was a linear trend in the effect of time.

## 3. Results

The response rates of the mailed surveys (n = 231) and emailed surveys (n = 290) to the previously established research database were 26% (n = 51) and 62.4% (n = 181), respectively. The total response rate was 46.3%. The majority of our participants were married (44%), were employed full time (48.3%), and had a higher education or university degree (83.1%). The survey respondents had high levels of psychological distress, and impaired quality of life (compared to normative data published previously), as presented in Table 1 (participant features). As shown in Table 2, most participants reported using at least one form of complementary therapy 12-months prior to study participation to treat their emotional or mental health problems (58.6%), eating disorders (16.8%), or to lose weight (31%). The most widely utilized complementary therapies were “talking about the problem” and “getting fit by doing more exercise”. Favorable attitudes towards complementary therapies and experiences of using them are reported in Table 2.

According to the results of our study, only 1.7% of the participants had attempted using an eating disorder related Medicare item. Approximately 87.1% of the participants were not aware of the existence of the eating disorder Medicare item or had not attempted to use it. These results are presented in Table 3.

Table 4 shows the influence of COVID-19 on mental health, eating disorders and quality of life. Less than half of the participants (46.1%) took the COVID-19 test. None of these participants tested positive for COVID-19. The in-person work-related activities of the participants were not affected by COVID-19 (41.8%). However, 30% of participants stopped attending other in-person activities, such as music and sporting events or lessons. The in-person social activities of the participants were affected to some extent (13.4%). COVID-19 did not affect the eating behavior of 51.3% of the participants. Only 5.6% of the participants were extremely worried about contracting the virus. The information delivery concerning COVID-19 by the government was rated quite clear and understandable by 38.8% of the participants.

Preliminary findings of the current study showed that the respondents had no significant changes in the levels of psychological distress measured via K10 (K10: 21.4 ± 9.8 vs. 19.0 ± 7.1, *p* < 0.171), and impaired quality of life (SF12: 50.9 ± 8.0 vs. 48.3 ± 9.5, *p* < 0.055) at 15-year follow-up time point. Eating disorder symptoms increased over time (EDE-Q global: 2.1 ± 1.4 vs. 2.9 ± 1.4, *p* < 0.013). Eating disorder symptom severity, including restraint, weight concern, shape concern and eating concerns, had all increased from baseline and reached their peak at year 15 when the COVID-19 pandemic was at its highest (*p* < 0.0001 for all). Psychological distress scores increased gradually from baseline to year 9 (2014), had a decline from year 9 to year 12 and reached their highest score after year 12 (*p* < 0.0001). Physical health-related quality of life decreased over time until year 12, after which there was an increase at year 15 (*p* < 0.0001). There was a similar trend for mental health related quality of life; however, the changes were smaller and non-significant (*p* = 0.158). See Figure 1, Figure 2, Figure 3, Figure 4, Figure 5 and Figure 6 below.

## 4. Discussion

In this current study of moderately symptomatic participants, we examined the effects of the COVID-19 pandemic on eating disorder symptoms and health related quality of life over 15 years in a symptomatic group of community dwelling participants. Overall, eating disorder symptom severity including restraint, weight concern, shape concern and eating concern increased since baseline at year one and reached its peak at year 15 (2020). This was noted when the COVID-19 pandemic was at its highest. In this study, we observed a decline in psychological distress score of the participants from year 9 to year 12. However, these scores then increased over time and reached their peak at year 15 follow-up. Physical health related quality of life decreased with time from baseline to year 12 and then increased again from year 12 to year 15. Mental health related quality of life followed the same trend; however, the changes were not significant.

The results of our study align with previous research conducted on the general population without any history of eating disorders, which has reported a significant decline in mental health during the COVID-19 pandemic. This decline was evident through the self-reported worsening of various symptoms, such as anxiety, sadness, anger, loneliness, and sleep problems, among others. [23,24,25,26,27,28]. However, some studies have also reported a slight decrease in the mental health problems faced by the general population after the ease of restrictions, compared to the initial negative impacts of the COVID-19 pandemic [29]. Despite these improvements, the changes did not reach the pre-pandemic levels of mental wellbeing [30]. Nonetheless, it is important to note that, despite the improvements, the current state of mental health among the general population still remains a matter of concern.

Similarly, eating disorder symptom severity increased post-pandemic, particularly in patients with eating disorders [31]. Further, Ramalho et al. (2022) found that psychological distress, measured by K10 questionnaire, mediated disordered eating behaviors in a cross-sectional study of adults during the first COVID-19 lockdown period [13]. However, there are some inconsistent findings, particularly in a clinical population [24], where it is possible that some people experience positive effects of confinement due to less exposure to societal judgment and criticism (e.g., from family, friends, and doctors). In addition, more recent longitudinal studies found a rapid decline in distress levels of patients with eating disorders soon after the first increase post-pandemic [32]. On the other hand, a recent systematic review by Devoe et al. (2023) showed increased hospital admission, increased eating disorder symptoms, anxiety and depression during the pandemic lockdown [10]. However, most studies included in this review failed to mention the effects of possible confounders and adjust for important covariates in their analysis, for example, there was a lack of adjustment based on gender or age. This emphasizes the importance of further examining the temporal changes in mental health and eating disorder symptoms of patients with eating disorders in the future.

The findings of the current study showed that the COVID-19 lockdown imposed significant changes on eating behaviors, physical health related quality of life and psychological distress. However, mental health related quality of life remained unchanged. This may be because our study is underpowered, or may reflect the complexity of the construct of mental health related quality of life. Mental health related quality of life measures the impact of poorer mental health on a person’s perceived quality of life, but this impact may be reduced where external factors are operating, and mental health related quality of life may be less sensitive to change than a specific measure of psychological distress. During the COVID-19 pandemic, there may have been a positive effect on quality of life because of reduced social interactions and social activities around food. For people with eating disorders, such social interactions may trigger social anxiety [33]. Nevertheless, our results emphasize the importance of designing interventions to alleviate the consequences of the COVID-19 pandemic on eating behaviors or eating disorders, creating a basis for upcoming situations involving lockdown and social isolation.

With regards to the question as to whether the COVID-19 pandemic may be more associated with increased severity and prevalence of eating disorders than with new cases (or incidence), we observed a peaking and rising trajectory in eating disorder symptoms. This was evident after the onset of the COVID-19 pandemic in both people with low, as well as those with high, pre-COVID-19 eating disorder symptom severity [34,35]. Notably, this was not due to increased availability of eating disorder services, as very few participants were aware of the existence of these services (under 3%).

### Strengths and Limitations

This longitudinal study aimed to investigate the effects of the COVID-19 pandemic on an eating disorder symptomatic cohort. Most notably, this study is also the first to report on public awareness of and use of the eating disorder related Medicare item in Australia. This study had some important limitations to consider when interpreting the results, including the use of self-reported surveys. These may be prone to reporting and recall biases. For the purpose of this study, we recruited a combination from the general population [22,36]. This cohort was representative of young woman who were living in the Australian Capital Territory, and was a convenience sample of higher education university and college students. The population was broadly representative of young women with at least a high school education in Australia [37,38,39]. The surveys were distributed to campuses with diverse socio-economic backgrounds. Although there is a very high proportion (56%) of the Australian population who engage in higher education [40], this sample was not representative of the wider Australian population. Therefore, the results may not be generalizable to other genders than females, to senior citizens, and to people with no eating disorder symptoms. The findings of the current study may provide useful information for future studies to examine, to a greater extent, issues concerning mental health, for example, differences in gender, age, and education. Such groups may be imported to target interventions aimed at improving mental health in the future.

## 5. Conclusions

The COVID-19 crisis has presented a number of challenges for individuals struggling with eating disorders. These challenges have the potential to worsen the symptoms of the disorder and increase the risk of developing or worsening disordered eating behaviors. For women in particular, the pandemic has exacerbated psychological distress and negatively impacted their physical health-related quality of life. The pandemic has resulted in significant changes to daily life for many people. Isolation, job loss, financial stress, and uncertainty about the future have all contributed to an increase in anxiety and depression. For individuals with eating disorders, these factors can trigger or worsen their symptoms. The disruption to daily routines, changes in food availability, and reduced access to healthcare services have also added to the challenge of managing eating disorders.

Women are particularly vulnerable to the negative effects of the pandemic on mental and physical health. Women are more likely to experience economic hardship, work in industries with high levels of exposure to the virus, and take on additional caregiving responsibilities. These stressors can further exacerbate the psychological distress associated with eating disorders and increase the risk of developing new symptoms. Overall, the COVID-19 crisis has highlighted the need for increased support and resources for individuals struggling with eating disorders, particularly for women, who may be disproportionately impacted. It is important for healthcare providers to be aware of the unique challenges faced by individuals with eating disorders during this time and to provide appropriate care and support.

Increasing community awareness of the existence and availability of the eating disorder Medicare item newly introduced by the Australian government may be beneficial for those affected by eating disorders in general and specifically during a crisis such as the COVID-19 pandemic. It is crucial for people with eating disorders to obtain support and have access to resources, even during these challenging times. Provision and access to interventions to support the people affected by eating disorders are a high priority, especially during these times and similar future situations. Overall, further research is needed to explore the impacts of COVID-19 and other pandemics on individuals with eating disorders. Such research should aim to identify effective strategies to reduce the negative impacts of pandemics on eating disorder patients and to improve their overall quality of life. For instance, research can explore how telehealth and online resources can be used to provide effective treatment and support to individuals with eating disorders during pandemics. Additionally, researchers can study the long-term impact of the pandemic on the development and course of eating disorders and identify ways to mitigate these effects. Furthermore, research can investigate how to address the unique challenges faced by marginalized communities, such as individuals with low income, people of color, and LGBTQ+ individuals, who may be disproportionately affected by the pandemic and its impact on eating disorders.

## Figures and Tables

**Figure 1 nutrients-15-01676-f001:**
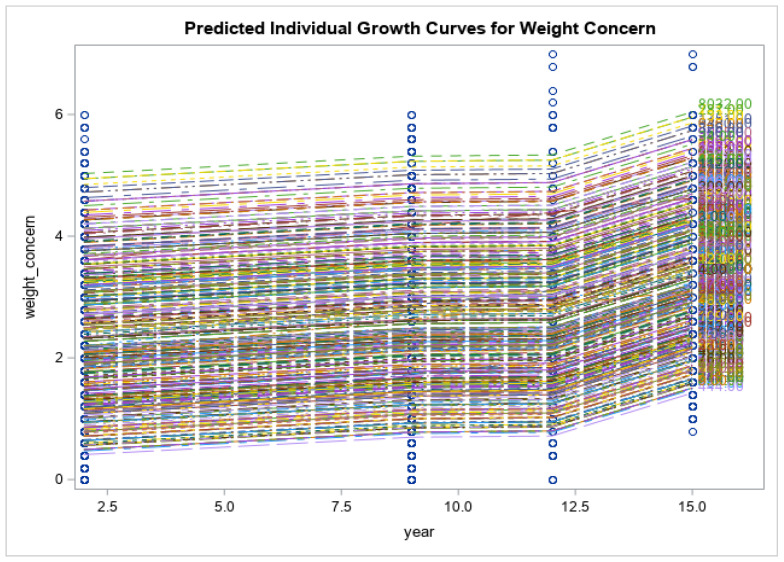
Predicted individual growth curves for weight concern. The blue lines indicate the measurement time points.

**Figure 2 nutrients-15-01676-f002:**
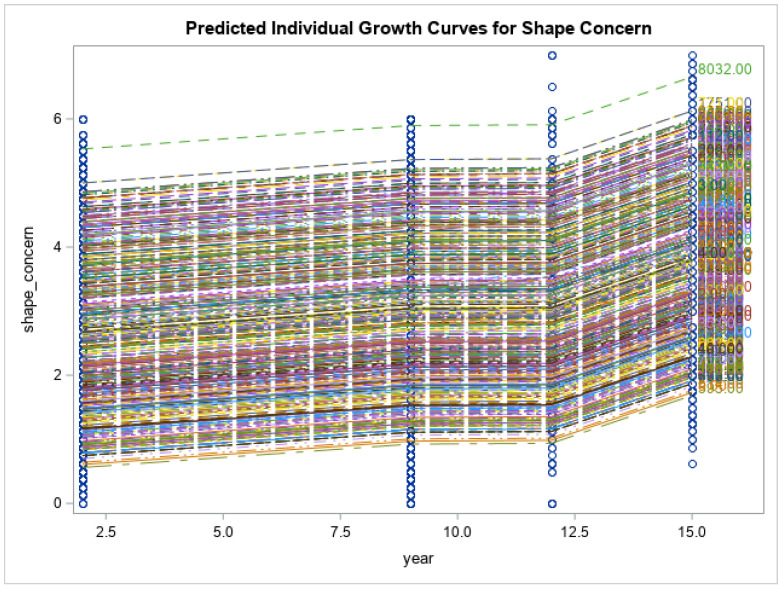
Predicted individual growth curves for shape concern. The blue lines indicate the measurement time points.

**Figure 3 nutrients-15-01676-f003:**
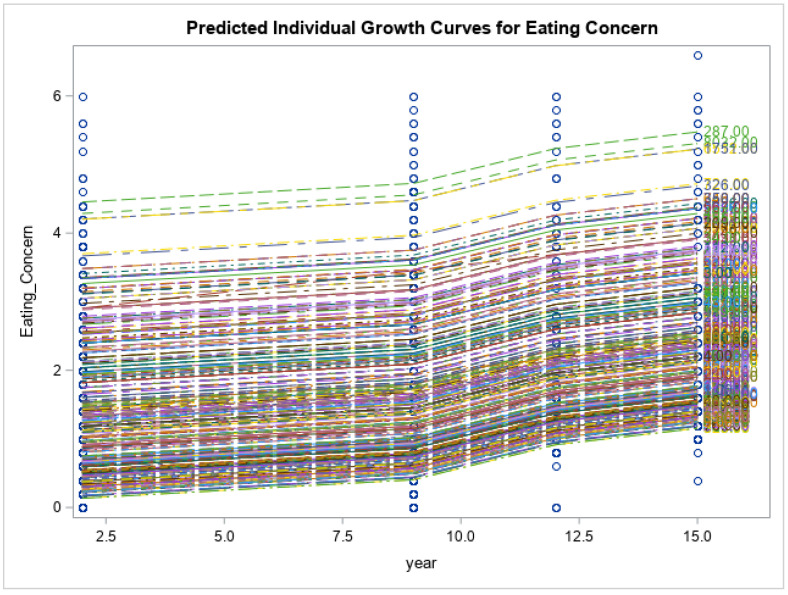
Predicted individual growth curves for eating concern. The blue lines indicate the measurement time points.

**Figure 4 nutrients-15-01676-f004:**
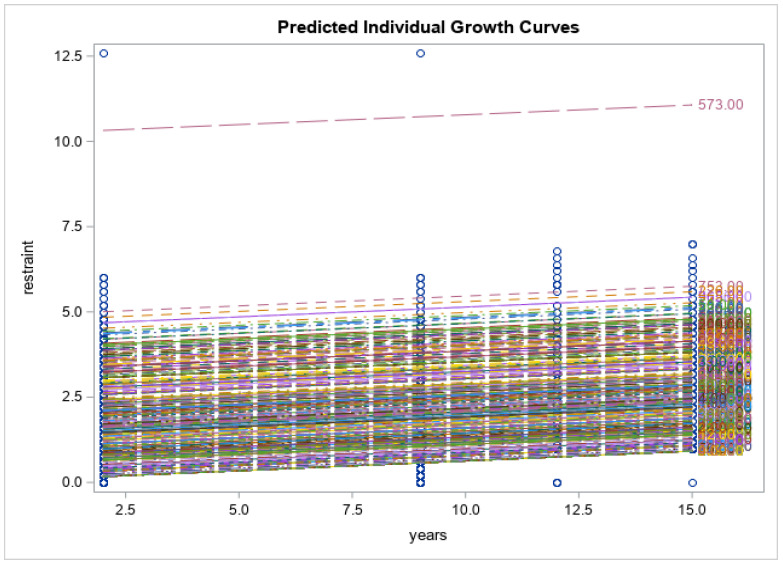
Predicted individual growth curves for eating restraint. The blue lines indicate the measurement time points.

**Figure 5 nutrients-15-01676-f005:**
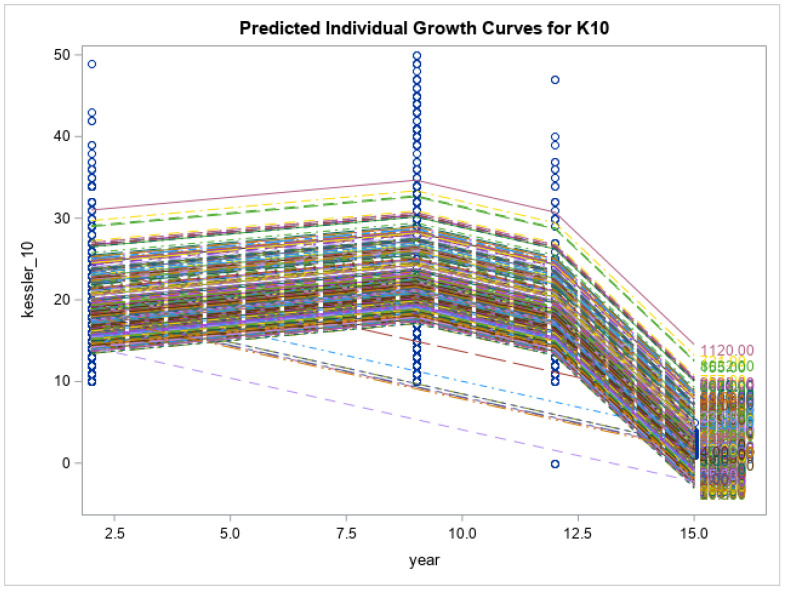
Predicted individual growth curves for K10. The blue lines indicate the measurement time points.

**Figure 6 nutrients-15-01676-f006:**
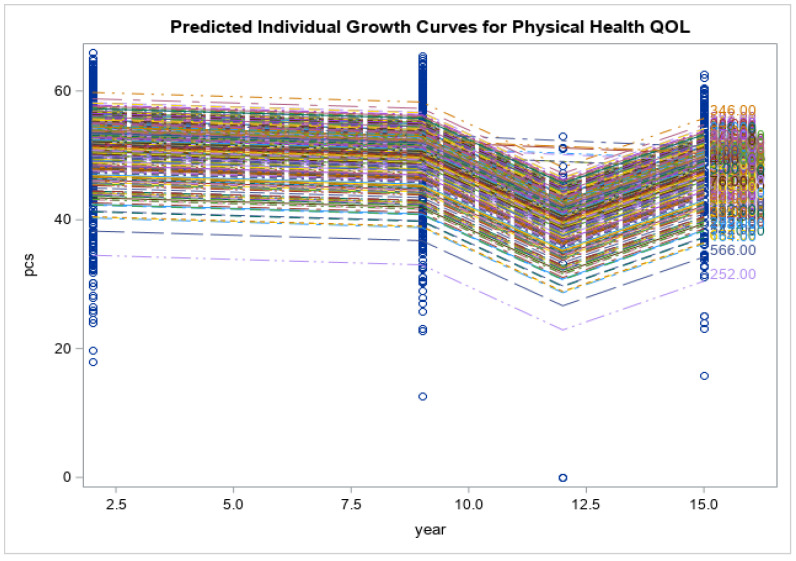
Predicted individual growth curves for physical health related. The blue lines indicate the measurement time points.

**Table 1 nutrients-15-01676-t001:** Participant’s demographic and health related features (n = 232).

Variables	N (%)
Marital status	
Single	34 (14.7)
Married	102 (44.0)
Living as married	47 (20.3)
Separated or divorced	33 (14.2)
Widowed	2 (0.9)
Occupation	
Employed fulltime	112 (48.3)
Employed Part time	51 (22)
Home maker ^	9 (3.9)
Student	6 (2.6)
Not employed, recovering, other	25 (10.8)
Education	
Year 10 & 12	15 (6.5)
Trade certificate	1 (0.4)
Undergraduate	9 (3.9)
University degree	193 (83.1)
Mean (SD), Median (Range), n	
BMI (kg/m^2^)	28.2 (6.7), 26.7 (39.01), 211
EDE-Q	
Restraint	2.9 (1.7), 2.8 (7.0), 209
Eating Concern	2.2 (1.4), 1.6 (6.20), 205
Weight concern	3.3 (1.6), 3.0 (6.20), 210
Shape concern	3.8 (1.7), 3.9 (6.25), 208
Global score	3.0 (1.4), 2.7 (6.04), 194
Psychological distress (Kessler 10)	19.1 (7.7), 17.0 (40.0), 208
Physical health related quality of life (SF-12)	48.4 (9.5), 51.7 (46.7), 203
Mental health related quality of life (SF-12)	45.5 (12), 50.4 (48.2), 203

Home maker: ^ a person who is not employed or student, nor looking for a job, and who is attending house duties in their home; SD: Standard Deviation; BMI: Body Mass Index; EDE-Q: Eating Disorder Examination Questionnaire [21], Kessler 10 [17] components of SF12, including physical wellbeing and mental wellbeing [19].

**Table 2 nutrients-15-01676-t002:** Treatments or activities within the past 12 months prior to study (N = 232).

	Emotional Ormental Health Problems	Eating Disorders	Lose Weight
	N (%)		
Used some type of treatment	136 (58.6)	39 (16.8)	72 (31)
Talked about the problem	110 (47.4)	17 (7.3)	11 (4.7)
Psychotherapy (focus on cause)	42 (18.1)	6 (2.6)	4 (1.7)
Psychotherapy (Cognitive Behaviour Therapy)	45 (19.4)	9 (3.9)	9 (3.9)
Psychotherapy (focus on current relationships)	38 (16.4)	3 (1.3)	2 (.9)
Alternative therapy (naturopathy, homeopathy, aromatherapy)	22 (9.5)	2 (.9)	5 (2.2)
Massage, spinal manipulation, or acupuncture	62 (26.7)	1 (.4)	3 (1.3)
Relaxation therapy (meditation, stress management, yoga)	72 (31)	6 (2.6)	7 (3.0)
Assertiveness or social skills training	9 (3.9)	1 (.4)	1 (.4)
Drinking alcohol to relax	60 (25.9)	3 (1.3)	3 (1.3)
Hypnosis	5 (2.2)	2 (.9)	3 (1.3)
Following a self-help treatment manual	25 (10.8)	6 (2.6)	9 (3.9)
Getting fit by increasing time spent on exercise	80 (34.5)	17 (7.3)	57 (24.6)
Getting out and about more or finding some new hobbies	60 (25.9)	5 (2.2)	8 (3.4)
Getting information about the problem and available services	45 (19.4)	5 (2.2)	9 (3.9)
Medication (valium, serapax)	25 (10.8)	3 (1.3)	3 (1.3)
Anti-depressant medication (Prozac or Zoloft)	42 (18.1)	4 (1.7)	4 (1.7)
Vitamins and/or minerals (folate, B group vitamins, amino acids etc)	55 (23.7)	5 2.2	5 (2.2)
Herbal medicines (Valerian, St John’s Wort, ginseng)	19 (8.2)	4 1.7	6 (2.6)

**Table 3 nutrients-15-01676-t003:** Eating disorder related Medicare item access (N = 232).

Medicare Item Use	N (%)
Attempted using the items	2 (0.9)
Accessed & used the items	4 (1.7)
No attempt/not aware of it	202 (87.1)

**Table 4 nutrients-15-01676-t004:** COVID-19′s influence on mental health, eating disorders and quality of life (N = 232).

	N (%)
Had COVID-19	0
Tested for COVID-19	107 (46.1)
Self isolated for 14 days	15 (6.5)
CVID-19′s effect on Work and/or study	
Stopped attending work and/or study in person	48 (20.7)
Continued work and/or study at home (online)	76 (32.8)
Stopped work and/or study all together	9 (3.9)
Continued work and/or study in person	97 (41.8)
COVID-19’s effect on other activities such as sport, music, clubs, etc.	
Stopped attending activities in person	71 (30.6)
Continued activities at home (online)	63 (27.2)
Stopped activities all together	17 (7.3)
Continued activities in person	54 (23.3)
COVID-19’s effect on social life	
Stopped socializing in person	31 (13.4)
Socializing via phone (online)	45 (19.4)
Stopped socializing all together (even online)	26 (11.2)
Continued socializing in person	87 (37.5)
COVID-19’s effect on household such as family and housemates	
One or more persons had COVID-19	1 (0.4)
One or more persons were unable to work	14 (6.0)
One or more persons are essential health worker	15 (5.2)
I am an essential health worker	62 (26.7)
None of the above	129 (55.6)
COVID-19’s effect on eating	
Not at all	119 (51.3)
More disordered eating (overeating, binge)	59 (25.4)
More dieting and/or exercise for weight control/loss	29 (12.5)
More weight/shape/eating concerns	39 (16.8)
Weeks unbale to attend work	
Not applicable	107 (46.1)
Unable to attend	98 (42.2)
Worried about contracting COVID-19	
Not at all	56 (24.1)
A little worried	81 (34.9)
Moderately worried	49 (21.1)
Very worried	11 (4.7)
Extremely worried	13 (5.6)
Worried about a family member contracting COVID-19	
Not at all	27 (11.6)
A little worried	79 (34.1)
Moderately worried	48 (20.7)
Very worried	39 (16.8)
Extremely worried	17 (7.3)
Rate the information delivery about COVID-19	
Very clear and understandable	35 (15.1)
Quite clear and understandable	90 (38.8)
Neutral	37 (15.9)
Quiet confusing	30 (12.9)
Very confusing	16 (6.9)

## Data Availability

The data presented in this study are available on request from the corresponding author. The data are not publicly available due to patients’ privacy and ethical permissions.

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
