# Peer review of "Community Women’s Lifestyle and Eating Disorders in the Era of COVID-19 Pandemic: A 15-Year Follow-Up Study"

_nutrients, 2023, doi:10.3390/nu15071676_

Round 1
Reviewer 1 Report
Please find my comments in the attached file.

Reviewer 2 Report
Dear Authors
Good synthesis in the introduction
Line 47 Please put the significance of SARS (severe acute respiratory syndrome)
Lines 56 to 59 should be clearer on the objectives and necessity of your study, clarify better the population you pretend studying (moderately ED symptomatic participants in 15 y follow up) and which innovative approach you pretend to do (see line 189 – 191)
In study design, line 193 to 198 should be ad to characterise the participant, in our opinion it is better at this place than at the end of the study in Strengths and Limitations.
Line 95 please put the significance of CTs
Discussion: what about Eating disorder related Medicare item access, the few uses despite increasing ED stress, is it due to the difficult access to the doctors / hospitals? In your discussion you should better highlight the new insight coming from your study, your discussion is not enough and the recommendations should be explained more in detail. You study deserve better discussion and recommendations and conclusions, even if we do not consider that they are control missing in some experiments, we will reconsider after major revision. For future reader we think that it will be usefull!
We whish you success in this reformulation
Author Response
Please see the attachment. please dis-regard the file titled: author-coverletter-27677042.v1.pdf
binder reviewer 2 is the correct file which includes the response letter and the updated manuscript.

Round 2
Reviewer 2 Report
Dear Authors,
we are satisfied by your reformulation and wish you success